# Silk Fibroin-Modified Liposome/Gene Editing System Knocks out the *PLK1* Gene to Suppress the Growth of Lung Cancer Cells

**DOI:** 10.3390/pharmaceutics15122756

**Published:** 2023-12-12

**Authors:** Peng Pan, Xueping Liu, Mengqi Fang, Shanlong Yang, Yadong Zhang, Mingzhong Li, Yu Liu

**Affiliations:** National Engineering Laboratory for Modern Silk, College of Textile and Clothing Engineering, Soochow University, Suzhou 215123, China; ppanpanpeng@stu.suda.edu.cn (P.P.); 20214015002@stu.suda.edu.cn (X.L.); 20214215051@suda.stu.edu.cn (M.F.); 20215215053@stu.suda.edu.cn (S.Y.); 20215215066@stu.suda.edu.cn (Y.Z.)

**Keywords:** silk fibroin, liposome, gene editing, lung cancer cells

## Abstract

Polo-like protein kinase 1 (PLK1) plays a key role in lung cancer cell mitosis. The knockout of *PLK1* gene by the CRISPR–Cas9 system can effectively inhibit the proliferation of tumor cells, but there is no suitable vector for in vivo delivery. In this study, CRISPR–Cas9 gene knockout plasmids encoding sgRNA, Cas9 and green fluorescent protein were constructed. Then, the plasmids were packaged with liposome (Lip) and cholesterol-modified *Antheraea pernyi* silk fibroin (CASF) to obtain the CASF/Lip/pDNA ternary complex. The CASF/Lip/pDNA complex was transfected into lung cancer cells A549 to investigate the transfection efficiency, the *PLK1* gene knockout effect and the inhibitory effect on lung cancer cells. The results showed that the transfection efficiency of the CASF/Lip/pDNA complex was significantly higher than that of the Lip/pDNA binary complex, and the expression of *PLK1* in cells transfected with CASF/Lip/pDNA complexes was significantly lower than that in cells transfected with Lip/pDNA complexes. The CASF/Lip/pDNA complex significantly increased the apoptosis rate and decreased the proliferation activity of lung cancer cells compared with Lip/pDNA complexes. The cytotoxicity of the complexes was evaluated by coculture with the human bronchial epithelial cells BEAS2B. The results showed that CASF/Lip/pDNA complexes exhibited lower cytotoxicity than Lip/pDNA complexes. The fibroin-modified liposome/*PLK1* gene knockout system not only effectively inhibited the growth of lung cancer cells but also showed no obvious toxicity to normal cells, showing potential for clinical application in lung cancer therapy.

## 1. Introduction

Lung cancer is the leading cause of cancer-related death worldwide due to its high incidence, aggressive late diagnosis, and lack of effective treatments [1]. Lung cancer is usually divided into small-cell lung cancer and non-small-cell lung cancer (NSCLC), and the latter accounts for more than 80% of lung cancer cases [2,3]. Molecular targeted therapy and immunotherapy for NSCLC have made significant progress in the past two decades, but the prognosis of NSCLC is not good, and it is still an incurable disease for most patients [4,5,6,7].

Polo-like kinase 1 (PLK1) plays an important role in the initiation, maintenance and completion of mitosis and is overexpressed in a variety of tumors, and this overexpression is associated with poor clinical outcomes [8,9,10]. The short palindromic repeats (CRISPR)-associated protein system 9 (CRISPR–Cas9), derived from the archaeal adaptive immune system, can edit genetic material in a specific sequence-dependent manner [11,12]. The plasmid containing the Cas9 gene and the gene encoding sgRNA can be introduced into target cells to form the CRISPR–Cas9 gene knockout system, which can knock out the *PLK1* gene of target cells, promote the apoptosis of cancer cells and inhibit the proliferation and migration of cancer cells [13,14].

The key challenge to achieve application is the efficient delivery of the CRISPR–Cas9 system into cells [15,16]. Viral vectors can effectively deliver the CRISPR–Cas9 system, but the limited packaging capacity of viral vectors and safety issues such as carcinogenicity and immunogenicity have seriously affected the clinical transformation of viral vectors [17,18,19]. The delivery of CRISPR–Cas9 by nonviral vectors such as liposomes, polymeric materials, gold nanoparticles and graphene oxide has shown great potential in terms of safety and packaging capacity [20,21,22]. Among them, liposomes are one of the most popular nonviral vectors for gene delivery and have the advantages of biodegradability, biocompatibility and high nucleic acid encapsulation efficiency [23,24].

However, cytotoxicity and low transfection efficiency remain challenges for the clinical application of liposomes. The phagocytosis of liposomes by macrophages will increase the level of intracellular reactive oxygen species and induce the apoptosis of macrophages [25,26,27]. In vivo studies have shown that lipids induce hepatotoxicity and inflammatory responses, as well as DNA damage in the lungs [28]. In addition, due to the limitation of the large size of CRISPR–Cas9, liposomes cannot effectively deliver the CRISPR–Cas9 system [29,30,31]. Therefore, the modification of liposomes to improve the gene transfection efficiency of liposomes, thereby reducing the dosage and cytotoxicity, is necessary for tumor gene therapy.

Silk fibroin is a natural protein derived from silk. Due to the higher content of acidic amino acids compared to basic amino acids, the surface of *Antheraea pernyi* fibroin is negatively charged under neutral conditions. Silk fibroin has good biocompatibility and biodegradability and has been applied in tissue engineering, drug delivery and gene delivery [32,33]. As a drug delivery system, silk fibroin-coated liposomes can improve the cellular uptake rate of small-molecule drugs [34,35]. Silk fibroin-modified polyethylenimine shows reduced cytotoxicity as a gene delivery vector [36,37,38]. In addition, a silk fibroin coating improved the cell adhesion ability of nanoparticles and enhanced the interaction with target cells [39,40].

Based on the advantages of silk fibroin, we hypothesized that silk fibroin modified by cholesterol could fuse with cationic liposomes, jointly package and compress CRISPR–Cas9 gene knockout plasmids to form a CASF/Lip/pDNA ternary complex. Cholesterol-modified silk fibroin could neutralize the positive charge on the surface of cationic liposomes and reduce toxicity to normal lung cells. Moreover, the spacing and balanced distribution of hydrophilic/hydrophobic fragments on silk fibroin could promote the interaction of the CASF/Lip/pDNA complex with the lung cancer cell membrane, enhance the uptake of the complex by lung cancer cells and improve the transfection efficiency. Compared with the Lip/pDNA binary complex, the CASF/Lip/pDNA ternary complex-formed CRISPR–Cas9 gene knockout system could more effectively knock out the *PLK1* gene and inhibit the proliferation of lung cancer cells.

To confirm this hypothesis, cholesterol-modified silk fibroin (CASF) was obtained by linking thiolcholesterol to the side chain of *Antheraea pernyi* silk fibroin (ASF) using the bifunctional reagent 3-(2-pyridine dimercapto) propionate N-hydroxysuccinimide ester (SPDP). The product was identified by FTIR and ^1^H-NMR. Then, CASF and Lipofectamine 3000 were used to package the plasmid containing the Cas9 gene and the gene coding sgRNA to obtain the CASF/Lip/pDNA ternary complex. In vitro, the effects of the CASF/Lip/pDNA ternary complex on gene transfection efficiency, cell proliferation and apoptosis were investigated by confocal laser microscopy and flow cytometry after transfecting lung cancer cells A549 and compared with the results of using the Lip/pDNA binary complex. The knockout effect of the CRISPR–Cas9 system on the *PLK1* gene was detected by Western blotting. Furthermore, a CASF/Lip/pDNA ternary complex was cocultured with human bronchial epithelioid cells BEAS2B to investigate the reduction in cytotoxicity of the CASF/Lip/pDNA ternary complex compared with the Lip/pDNA binary complex. To investigate whether cholesterol-modified silk fibroin could fuse more efficiently with liposomes, reduce cytotoxicity and improve gene transfection efficiency, parallel experiments were performed with the ASF/Lip/pDNA as a control complex, which was prepared by packaging plasmids with unmodified silk fibroin and Lipofectamine 3000.

## 2. Materials and Methods

### 2.1. Preparation of ASF

As described previously [41], *Antheraea pernyi* silk (Dandong, China) was degummed three times with 2.5 g/L sodium carbonate solution (Sinopharm Chemical Reagent Co., Ltd., Shanghai, China) at 98 to 100 °C for 30 to 45 min. Then, it was rinsed thoroughly with deionized water and dried in an oven at 60 ± 2 °C. Next, 10 g of dried fibers were dissolved in 100 mL of melted Ca(NO_3_)_2_·4H_2_O (Sinopharm Chemical Reagent Co., Ltd., Shanghai, China) solutions at 105 ± 2 °C for 5 h. The cooled mixed solution was dialyzed in a dialysis bag (MWCO, 8–14 kDa) against deionized water for 2 days to obtain the regenerated *Antheraea pernyi* silk fibroin (ASF) solution. The fresh solution was then filtered using a 0.22 μm pore size filter (Millipore, Billerica, MA, USA).

### 2.2. Synthesis of CASF

N-succinimidyl-3-(2-pyridyldithio) propionate (SPDP, Sigma-Aldrich, Saint Louis, MO, USA) was dissolved in dimethyl sulfoxide. Then, 0.2 mL of SPDP (1 mg/mL) solution was added to 30 mL of ASF solution (1 mg/mL), and the reactants were stirred slowly in an ice bath for 8 h. The mixed solution was dialyzed with deionized water for 24 h at 4 °C to obtain the SPDP-activated ASF solution. Thiocholesterol (Sigma-Aldrich, Saint Louis, MO, USA) was dissolved in dimethyl sulfoxide, and then 0.2 mL of thiocholesterol (1 mg/mL) was added to the activated ASF solution and slowly stirred for 8 h in an ice bath. The mixed solution was dialyzed in cellulose (MWCO, 8–14 kDa) against deionized water for 24 h at 4 °C to obtain the CASF solution. At the end of dialysis, the solution was filtered using a 0.22 μm pore size filter.

### 2.3. Characterization of CASF

The Fourier transform infrared spectroscopy (FTIR) spectra of the ASF and CASF were obtained using a Nicolet iS5 spectrometer (Thermo Company, Madison, WI, USA). The measurements were performed with 0.8 cm^−^^1^ resolution in the range of 400–4000 cm^−1^.

Freeze-dried ASF and CASF (modified with cholesterol against ASF) were dissolved in D_2_O. Then 0.5 mL of the solution was transferred into a nuclear magnetic tube. A superconducting nuclear magnetic resonance (NMR) spectrometer (AVANCE III HD 400 MHz; Bruker, Billerica, MA, USA) was used to determine the ^1^H-NMR spectra.

### 2.4. Plasmid DNA Production

Based on the *PLK1* gene sequence (NM_005030.6 RefSeqGene, NCBI), we designed the DNA sequence encoding the sgRNA (5′-CACCGGAATCCTACGACGTGCTGGT-3′ (forward) and 5′-ACCAGCACGTCGTAGGATTCC-3′ (reverse)). The DNA sequence encoding the sgRNA was cloned into the pSpCas9(BB)-2A-GFP (PX458, Addgene, Cambridge, MA, USA) plasmid encoding Cas9 and green fluorescent protein (GFP) using BbsI enzyme (Thermo Fisher Scientific, Waltham, MA, USA). Then, the CRISPR–Cas9 gene knockout plasmid was propagated in *Escherichia coli* DH5α cells (Invitrogen, Carlsbad, CA, USA). Ultrapure, endotoxin-free plasmid pDNA was prepared using an Endo-free Plasmid Mini Kit (Omega Bio-Tek, Norcross, GA, USA) according to the manufacturer’s instructions. The plasmid concentration and purity were measured by a Nanodrop 2000 UV spectrophotometer (Thermo Fisher Scientific, Waltham, MA, USA).

### 2.5. Preparation of CASF/Lip/pDNA Complex

The Lip/pDNA binary complex was prepared using Lipofectamine™ 3000 Reagent (Thermo Fisher Scientific, Carlsbad, CA, USA) according to the manufacturer’s instructions. Briefly, 3 µL of Lipofectamine 3000 (1 µg/µL) was added to 50 µL of Opti-MEM (Thermo Fisher Scientific, Grand Island, NY, USA) and mixed. In another tube, 1 µg of pDNA (500 ng/µL) and 2 µL of P3000 reagent (1 µg/µL, Thermo Fisher Scientific, Carlsbad, CA, USA) were added to 50 µL of Opti-MEM and mixed. The contents of the two tubes were then combined by gentle pipetting and incubated for 15 min at room temperature to allow the formation of the Lip/pDNA (0/3/1) binary complex.

The CASF/Lip/pDNA (1/3/1) ternary complex was prepared by mixing 1 μL of CASF (1 µg/µL) with Lip/pDNA complexes, followed by vortexing for 15 s and incubation for 20 min at room temperature. The preparation method of the ternary complex CASF/Lip/pDNA (3/3/1, 5/3/1, 7/3/1, 9/3/1) and control complex ASF/Lip/pDNA (1/3/1, 3/3/1, 5/3/1, 7/3/1, 9/3/1) was the same as that of CASF/Lip/pDNA (1/3/1), except that the amount of CASF or ASF mixed was different.

### 2.6. Characterization of the CASF/Lip/pDNA Complex

The particle sizes and zeta potentials of the Lip/pDNA, CASF/Lip/pDNA and ASF/Lip/pDNA complexes were determined at 25 °C by utilizing a Zetasizer Nano-ZS90 analyzer (Malvern Panalytical, Malvern, UK).

The morphologies of the Lip/pDNA, CASF/Lip/pDNA and ASF/Lip/pDNA complexes were observed by scanning electron microscopy (SEM, Hitachi S-8100, Tokyo, Japan). The suspension of the complex was dropped on the silicon chip and dried at room temperature. The dried samples were sputter coated with gold for 90 s and examined using SEM. The particle size of the complexes in SEM images was measured using Image-J (1.51J8) software, at least 100 complexes were measured in each group of samples.

### 2.7. Gene Transfection of A549 Cells

Human lung adenocarcinoma cells A549 were purchased from ATCC (Manassas, VA, USA). A549 cells were cultured in Ham’s F–12K (Procell Life Science, Wuhan, China) containing 10% fetal bovine serum (FBS, Gibco, Waltham, MA, USA). Cells were grown in a polystyrene culture flask (NEST, Wuxi, China) using a humidified 5% CO_2_ incubator at 37 °C.

A549 cells were seeded on 12-well plates at a density of 2 × 10^5^ and cultured for 12 h. The Lip/pDNA, CASF/Lip/pDNA and ASF/Lip/pDNA complexes were added to the cell culture medium, and after culturing for 24 h, the expression of green fluorescent protein (GFP) in cells was detected by confocal laser microscopy (FV10, Olympus, Tokyo, Japan).

Levels of GFP expression were quantitatively determined by flow cytometry analysis. At 24 h post transfection, the cells were trypsinized, washed and measured by a flow cytometer (FC500, Beckman-Coulter, Brea, CA, USA). Untransfected cells were used as negative controls.

### 2.8. Western Blot Assay of PLK1 Gene Expression

At 48 h post transfection, cells were collected and lysed in 200 μL of lysis buffer. By 12% sulfate–polyacrylamide gel electrophoresis to separate the total cell lysates and blotted with rabbit anti-human PLK1 (ab189139, Abcam, Cambridge, UK) and anti-β-actin (20536-1-AP, Proteintech, Los Angeles, CA, USA). After incubation with DyLight fluorescent dye-conjugated secondary antibodies (A23720, Abbkine, Wuhan, China), images were obtained using the Odyssey CLx Imaging System (LI-COR Biosciences, Lincoln, NE, USA).

### 2.9. Apoptosis Assay of A549 Cells

Cell apoptosis was analyzed by flow cytometry according to the Annexin V-PE/7-AAD Kit (Yeasen, Shanghai, China). A549 cells were harvested 48 h after transfection. Then, 100 μL of the cell suspension was transferred to a flow tube, followed by 5 μL of Annexin V/PE and 10 μL of 7-ADD, and incubated in the dark for 15 min. Apoptosis was assessed by a flow cytometer, and 1 × 10^4^ cells per sample were examined. Untransfected cells were used as negative controls.

### 2.10. Viability of A549 Cells

A549 cells were seeded in 96-well culture plates, 1 × 10^4^ cells per well. After transfection of A549 cells for 24, 48 and 72 h, the medium was removed, and 90 μL of DMEM and 10 μL of CCK-8 were added. After incubation for 2 h, the optical density (OD) value was measured at 450 nm by using a microplate reader (TECAN-Spark, Mnnedorf, Switzerland). The cell viability (%) was calculated according to Formula (1). Untransfected cells were used as a control, and each experiment was performed in triplicate.
(1)$$\mathrm{Cell}\;\mathrm{viability}=\frac{{OD}_{sample}}{{OD}_{control}}\times 100\%$$

### 2.11. Cytotoxicity to BEAS2B Cells

Human bronchial epithelioid cells BEAS2B were purchased from ATCC (Manassas, VA, USA). BEAS2B cells were cultured in Dulbecco’s modified Eagle medium (DMEM, Gibco, Waltham, MA, USA) containing 10% fetal bovine serum. Cells in the ordinary course of polystyrene culture bottle (NEST, Wuxi, China) growth. Other culture conditions were the same as for A549 cells. BEAS2B cells were cocultured with the complex in the same way as A549 cells were transfected with the complex. After 24 h of coculturing BEAS2B cells with the complex, cell viability was measured by the same method as for A549 cells.

### 2.12. Statistical Analysis

Data are expressed as the mean ± standard deviation (SD). Statistically significant differences between groups were tested by one-way analysis of variance. Differences at *p* < 0.05 were considered statistically significant.

## 3. Results

### 3.1. Characterization of CASF

Silk fibroin and thiolcholesterol were cross linked by 3-(2-pyridine-dithiol) propionate N-hydroxysuccinimidyl ester (SPDP). As shown in Figure 1A, the NHS ester group on SPDP undergoes amidation with the amino group on the side chain of ASF, such that SPDP is covalently linked to ASF via an amide bond. The sulfhydryl group of thiolcholesterol was used to react with the dithiopyridine group of SPDP so that the thiolcholesterol was coupled to the ASF side chain in the form of a disulfide bond to obtain CASF.

Figure 1B shows the FTIR spectra of CASF. Absorption peaks of amide II (N-H bending vibration and C-H stretching vibration) and amide III (C-N stretching vibration and N-H bending vibration) of ASF appear at 1540 cm^−1^ and 1240 cm^−1^ [42,43]. The absorption peaks of amide II and amide III of CASF also appear at 1540 cm^−1^ and 1240 cm^−1^. Thiocholesterol has the characteristic absorption peak of the sulfhydryl group at 801 cm^−1^. After the silk fibroin was modified with cholesterol, there was no absorption peak at 801 cm^−1^, demonstrating that sulfhydryl groups were involved in the reaction and consumed. And the characteristic absorption of ASF at 1640 cm^−1^ shifted to 1650 cm^−1^ in CASF, suggesting that the NHS ester group on SPDP forms a new amide bond with the amino group on the ASF side chain. CASF shows a new absorption peak at 539 cm^−1^, which is the characteristic absorption peak of disulfide bonds [44], showing that disulfide bonds are formed between thiocholesterol and ASF via SPDP.

Figure 1C shows the ^1^H-NMR spectra of ASF and CASF. The chemical shift of 1 (δ, ~4.11 ppm) corresponds to the chemical shift of the proton in the amide bond (–NH–C(=O) –), 2 (δ, ~3.84 ppm) to –CH– in Ser, and 3 (δ, ~1.32 ppm) to the –CH_2_–NH_2_CH_3_ proton in Ala, all of which appeared on the ASF and CASF spectra [45]. The chemical shift of 4 (δ, 1.46~1.57 ppm) corresponds to the chemical shift of the –CH_2_– proton in the cholesterol [46]. A new proton peak at 5 (δ, ~2.75 ppm) appeared in the ^1^H-NMR spectrum of CASF, which is the chemical shift of the proton of the –S–S–CH_2_– structure [47]. Thiocholesterol is coupled to ASF via disulfide bonds mediated by SPDP, thus generating a new proton peak at 2.75 ppm. The results of the FTIR spectra and ^1^H-NMR spectra jointly suggested the successful modification of ASF with cholesterol.

### 3.2. Characterization of CASF/Lip/pDNA Complexes

CASF/Lip/pDNA ternary complex was prepared by fusing cholesterol-modified silk fibroin (CASF) with a Lip/pDNA binary complex. As shown in Figure 2A, cationic liposomes (Lip) and negatively charged pDNA form a Lip/pDNA binary complex by electrostatic interaction. Small granular liposomes compress pDNA to form larger aggregates (Lip/pDNA), with pDNA inside and liposomes outside [48]. The negatively charged CASF binds to the Lip/pDNA complex through electrostatic interactions, while the cholesterol on the side chain of CASF may be able to insert/fuse to the lipid molecules of the liposomes by hydrophobic forces. Finally, CASF fuses with the Lip/pDNA binary complex through electrostatic interactions and hydrophobic interactions to form a CASF/Lip/pDNA ternary complex.

Figure 2B shows the zeta potential of the CASF/Lip/pDNA ternary complex and the ASF/Lip/pDNA complex (control group). The zeta potential of the binary complex Lip/pDNA (0/3/1) was approximately + 37 mV. As the mass ratio of ASF and CASF increased, the zeta potential of the CASF/Lip/pDNA and ASF/Lip/pDNA complexes first decreased and then stabilized. The zeta potential of the CASF/Lip/pDNA ternary complex was +30 mV and +23 mV at CASF/Lip/pDNA mass ratios of 1/3/1 and 3/3/1, respectively. The zeta potential of CASF/Lip/pDNA complexes with CASF/Lip/pDNA mass ratios of 5/3/1, 7/3/1 and 9/3/1 did not change significantly compared with CASF/Lip/pDNA (3/3/1) complexes. The zeta potentials of ASF/Lip/pDNA complexes with mass ratios of 1/3/1, 3/3/1 and 5/3/1 were +25 mV, +20 mV and +18 mV, respectively. The zeta potential of the ASF/Lip/pDNA complexes with mass ratios of 7/3/1 and 9/3/1 did not change significantly compared to the ASF/Lip/pDNA complexes with mass ratios of 5/3/1. The zeta potential of the CASF/Lip/pDNA ternary complex and the ASF/Lip/pDNA complex showed that both CASF and ASF were effective in neutralizing the positive charge on the surface of Lip/pDNA.

Figure 2C shows the particle sizes of the CASF/Lip/pDNA ternary complex and the ASF/Lip/pDNA complex. The particle size of the binary complex Lip/pDNA (0/3/1) was approximately 70 nm. The particle size of the CASF/Lip/pDNA ternary complexes increased with increasing CASF mass ratio. The particle sizes of CASF/Lip/pDNA complexes with mass ratios of 1/3/1, 3/3/1, 5/3/1, 7/3/1 and 9/3/1 were 113 nm, 147 nm, 158 nm, 191 nm and 301 nm, respectively. The particle size of the ASF/Lip/pDNA complex increased first and then stabilized with an increasing mass ratio of silk fibroin. The particle sizes of ASF/Lip/pDNA complexes with mass ratios of 1/3/1, 3/3/1, 5/3/1, 7/3/1 and 9/3/1 were 98 nm, 118 nm, 132 nm, 145 nm and 147 nm, respectively. The particle size change of the CASF/Lip/pDNA ternary complex was more pronounced than that of the ASF/Lip/pDNA complex, showing that the liposome surface could bind more CASF (compared to ASF).

The morphologies of the CASF/Lip/pDNA complexes and ASF/Lip/pDNA complexes were observed by scanning electron microscopy. As shown in Figure 2D, the Lip/pDNA, CASF/Lip/pDNA and ASF/Lip/pDNA complexes were spherical particles. The particle sizes of the Lip/pDNA, CASF/Lip/pDNA with mass ratios of 1/3/1 and 3/3/1 and ASF/Lip/pDNA with mass ratios of 1/3/1 and 3/3/1 complex were approximately 30, 40, 50, 40, and 40 nm, respectively. The particle size of the complexes in the dry state was smaller than that of the complexes in the hydrated state in Figure 2C.

### 3.3. Transfection of Lung Cancer Cells A549 with CASF/Lip/pDNA Complexes

A549 cells were transfected with Lip/pDNA (0/3/1), CASF/Lip/pDNA and ASF/Lip/pDNA complexes in the presence of serum. As shown in Figure 3A, the untransfected cells did not express green fluorescent protein. Green fluorescence was observed in A549 cells after transfection of the Lip/pDNA (0/3/1) complex. Compared with that of the Lip/pDNA (0/3/1) complex, the green fluorescence intensity of the CASF/Lip/pDNA complexes with mass ratios of 1/3/1 and 3/3/1 and ASF/Lip/pDNA complexes with mass ratios of 1/3/1 was significantly increased. Bright-field images showed that the untransfected A549 cells (NC) were fully diffused and spindle shaped, with only a few cells spherical. After transfection with the Lip/pDNA (0/3/1) complex, most A549 cells were spindle shaped, and a few cells were spherical. Compared with the transfection of A549 cells with Lip/pDNA (0/3/1) complex, after transfection of cells with CASF/Lip/pDNA (1/3/1, 3/3/1) and ASF/Lip/pDNA (1/3/1, 3/3/1) complexes, the number of A549 cells with spherical shape increased significantly, the connections between cells decreased, the distance between cells increased, and the number of non-adherent cells increased. The results showed that CASF/Lip/pDNA and ASF/Lip/pDNA complexes had stronger growth inhibition effects on lung cancer cell A549 than Lip/pDNA complexes.

Transfection efficiency was obtained by counting the number of GFP-positive cells using flow cytometry (Figure 3B). The percentage of GFP-positive cells in the NC group was 1.89%, and the transfection efficiency of the Lip/pDNA (0/3/1) complex was 26.07%. The transfection efficiency of CASF/Lip/pDNA complexes with mass ratios of 1/3/1 and 3/3/1 and ASF/Lip/pDNA complexes with mass ratios of 1/3/1 and 3/3/1 was 35.97%, 33.73%, 33.30% and 27.43%, respectively. Transfection efficiency was significantly increased in CASF/Lip/pDNA complexes (1/3/1, 3/3/1) and ASF/Lip/pDNA (1/3/1) complexes compared to Lip/pDNA complexes (0/3/1) (Figure 3C, *p* < 0.01). The transfection efficiency of the ASF/Lip/pDNA complex (3/3/1) did not change significantly compared with that of the Lip/pDNA (0/3/1) complex (Figure 3C). The transfection efficiency of the CASF/Lip/pDNA complex was higher than that of the ASF/Lip/pDNA complex at the same mass ratio. This suggests that CASF can improve the transfection efficiency of Lip/pDNA complexes more than ASF.

The effect of *PLK1* gene knockout was detected by Western blotting. As shown in Figure 3D, after transfection with Lip/pDNA (0/3/1), CASF/Lip/pDNA (1/3/1, 3/3/1) complexes and ASF/Lip/pDNA (1/3/1, 3/3/1) complexes, the Western blot bands of PLK1 protein became lighter (compared with the NC group). This suggested that the transfection complex was able to reduce PLK1 protein expression. The CASF/Lip/pDNA (1/3/1) complex showed the shallowest band, showing the best knockdown effect, which was consistent with the transfection efficiency results.

### 3.4. Apoptosis Rate and Proliferation Activity of A549 Cells

The apoptotic rates of A549 cells transfected with different complexes were counted using flow cytometry (Figure 4A). The apoptosis rate of untransfected A549 cells (NC) was 2.60%. The apoptosis rates of A549 cells transfected with Lip/pDNA, CASF/Lip/pDNA complexes with mass ratios of 1/3/1 and 3/3/1 and ASF/Lip/pDNA complexes with mass ratios of 1/3/1 and 3/3/1 were 20.32%, 30.13%, 27.28%, 25.13% and 21.80%, respectively. Figure 4B shows that the apoptosis rate of A549 cells transfected with CASF/Lip/pDNA (1/3/1, 3/3/1) complexes was significantly higher than that of cells transfected with Lip/pDNA complexes (*p* < 0.001). These results indicated that CASF-modified liposomes significantly enhanced the apoptosis-inducing ability of Lip/pDNA complexes in A549 cells. The apoptosis rate of A549 cells in the CASF/Lip/pDNA ternary complex group was higher than that in the control ASF/Lip/pDNA complex group, indicating that the CASF/Lip/pDNA ternary complex more effectively promoted tumor cell apoptosis.

Figure 4C illustrates the cell viability of A549 cells transfected with different complexes. The cell viability of the untransfected complex was 100%. After 24 h of transfection, the cell viabilities of Lip/pDNA (0/3/1), CASF/Lip/pDNA complexes with mass ratios of 1/3/1 and 3/3/1 and ASF/Lip/pDNA complexes with mass ratios of 1/3/1 and 3/3/1 were 28.3%, 27.97%, 30.29%, 29.65% and 41.55%, respectively. At 48 h after transfection, cell viability of CASF/Lip/pDNA complexes with mass ratios of 1/3/1 (12.19%) was significantly decreased compared with Lip/pDNA (13.73%) (*p* < 0.05). The cell viability of the CASF/Lip/pDNA complex with mass ratios of 3/3/1 and ASF/Lip/pDNA with mass ratios of 1/3/1 and 3/3/1 was 13.80%, 16.06% and 25.90%, respectively, which was not significantly lower than that of the Lip/pDNA group. After 72 h of transfection, the cell viabilities of Lip/pDNA (0/3/1), CASF/Lip/pDNA complexes with mass ratios of 1/3/1 and 3/3/1 and ASF/Lip/pDNA complexes with mass ratios of 1/3/1 and 3/3/1 were 19.47%, 9.21%, 13.71%, 10.17% and 19.26%, respectively. Compared with the Lip/pDNA group, the cell viability of the CASF/Lip/pDNA (1/3/1, 3/3/1) groups was significantly decreased (*p* < 0.001). These results demonstrated that CASF modification enhanced the inhibitory effect of Lip/pDNA complexes on lung cancer cells. The viability of A549 cells in the CASF/Lip/pDNA ternary complex group was lower than that of the ASF/Lip/pDNA complex with the same mass ratio, indicating that the CASF/Lip/pDNA ternary complex had a stronger ability to inhibit the proliferation of A549 cells than the ASF/Lip/pDNA complex.

### 3.5. Transfection of Normal Lung Cells BEAS2B with CASF/Lip/pDNA Complexes

The expression of GFP in BEAS2B cells was observed by fluorescence microscopy 24 h after coculture. As shown in Figure 5A, cells that were not cocultured with the complex showed no green fluorescence. After coculture with the Lip/pDNA (0/3/1) complex, a small number of BEAS2B cells showed weak green fluorescence. The number of fluorescent cells cocultured with CASF/Lip/pDNA (1/3/1, 3/3/1) and ASF/Lip/pDNA (1/3/1, 3/3/1) complexes was lower than that of Lip/pDNA (0/3/1) complexes. Bright-field images showed that the BEAS2B cells not cocultured with the complex (NC) were fully spread, and the cells were spindle-shaped or polygonal. After being cocultured with the Lip/pDNA (0/3/1) complex, part of the cells shrank and became round, and the intercellular space became larger. Cells cocultured with CASF/Lip/pDNA (1/3/1, 3/3/1) or ASF/Lip/pDNA (1/3/1, 3/3/1) were fusiform or polygonal, and only a few cells were round. CASF/Lip/pDNA and ASF/Lip/pDNA complexes had less effect on cell morphology than Lip/pDNA complexes, suggesting that CASF could reduce the cytotoxicity of Lip/pDNA.

To assess the effects of CASF and ASF modification on the cytotoxicity of Lip/pDNA complexes, BEAS2B cell proliferation viability was measured by CCK-8 assay. As shown in Figure 5B, cell viability was 66.87% after coculture with the Lip/pDNA (0/3/1) complex. The viability of BEAS2B cells cocultured with CASF/Lip/pDNA complexes with mass ratios of 1/3/1 and 3/3/1 and ASF/Lip/pDNA complexes with mass ratios of 1/3/1 and 3/3/1 for 24 h was 80.50%, 82.18%, 77.16% and 84.58%, respectively. Compared with the Lip/pDNA (0/3/1) complex, BEAS2B cells cocultured with the CASF/Lip/pDNA (1/3/1, 3/3/1) complex and ASF/Lip/pDNA (1/3/1, 3/3/1) complex had higher viability (*p* < 0.05). The results showed that the CASF and ASF modifications reduced the cytotoxicity of the liposomes, which was consistent with the bright-field photographs of the cells.

## 4. Discussion

Introducing foreign genes into tumor cells to regulate protein expression is an effective way to inhibit the growth of tumor cells. [49,50]. In the process of introducing foreign genes into cells, gene delivery carriers play a key role. A liposome is one of the most widely used non-viral vectors, and a large number of studies have used them for cancer gene therapy [51,52,53]. However, the toxicity of liposomes to normal tissues and low transfection efficiency limit the clinical application of liposomes [29,30,31]. In this study, the CASF/Lip/pDNA ternary complex was prepared by the fusion of CASF with a Lip/pDNA binary complex. The surface zeta potential of CASF/Lip/pDNA was significantly lower than that of Lip/pDNA. Transfection experiments showed that the transfection efficiency of the CASF/Lip/pDNA ternary complex was significantly higher than that of the Lip/pDNA binary complex. Apoptosis and proliferation assays showed that the CASF/Lip/pDNA ternary complex enhanced the inhibitory effect on A549 cells compared with the Lip/pDNA binary complex. The results of experiments with transfected normal lung cells showed that the cytotoxicity of the CASF/Lip/pDNA ternary complex was significantly reduced in normal lung cells BEAS2B compared with the Lip/pDNA binary complex. This evidence suggests that the CASF-modified liposome/*PLK1* gene knockout system can effectively improve the transfection efficiency and enhance the inhibitory ability of lung cancer cells while reducing toxicity to normal lung cells.

Silk fibroin has a negative charge in a neutral environment [54,55] and can neutralize part of the positive surface charge of Lip/pDNA complexes. The contents of Arg, His and Lys in *Antheraea pernyi* silk fibroin are approximately 2.59%, 0.80% and 0.07%, respectively [56], and the free amino group or imino group on their side chains is coupled to cholesterol to produce CASF (Figure 1A). *Antheraea pernyi* silk fibroin basically does not contain Cys or other molecules containing free sulfhydryl groups [57]. The zeta potential of the complexes was significantly decreased after modification of the liposomes with CASF and ASF (Figure 2B), suggesting that CASF and ASF effectively neutralized the positive charge on the surface of the liposomes. The surface zeta potential of the CASF/Lip/pDNA complex was higher than that of the ASF/Lip/pDNA complex at the same mass ratio. The possible reason is that CASF binds to liposomes mainly through cholesterol on the side chain, while ASF binds to liposomes mainly through electrostatic interactions, which neutralize more positive charges. The particle size of the CASF/Lip/pDNA ternary complex was larger than that of the control ASF/Lip/pDNA complex with the same mass ratio. The possible reason is that the binding of CASF to the liposome surface through electrostatic and hydrophobic forces makes the fusion degree of CASF to liposomes higher than that of ASF so that more silk fibroin can be bound to the liposome surface.

The size, surface zeta potential and shape of the transfection complex have significant effects on transfection efficiency [48]. CASF binds to the surface of Lip and changes the surface potential, size and surface structure of the Lip/pDNA complex, thus affecting the transfection efficiency of the Lip/pDNA complex. CASF can improve the transfection efficiency of the Lip/pDNA binary complex in the presence of serum. After transfection of A549 cells for 24 h, the CASF/Lip/pDNA (1/3/1, 3/3/1) and ASF/Lip/pDNA (1/3/1) complexes showed higher transfection efficiency than the Lip/pDNA complexes (Figure 4B). There are two possible mechanisms. On the one hand, the RGD (Arg-Gly-Asp) sequence of CASF can promote the interaction of the CASF/Lip/pDNA complex with the lung cancer cell membrane, enhance the uptake of the complex by lung cancer cells and improve transfection efficiency [58]. On the other hand, ASF itself has a negative charge, which reduces the nonspecific adsorption of serum proteins by liposomes and improves the stability of the transfection complex. The transfection efficiency of the CASF/Lip/pDNA ternary complex was higher than that of the ASF/Lip/pDNA complex, probably because CASF was more able to fuse with liposomes, which enhanced the interaction between the CASF/Lip/pDNA complex and cell membrane.

CASF can enhance the inhibitory effect of the Lip/pDNA binary complex on lung cancer cells A549. PLK1 is a key protein in mitosis, and knocking out the *PLK1* gene can promote cell apoptosis and inhibit cell proliferation [13,14]. Western blot experiments showed that transfection of CASF/Lip/pDNA (1/3/1, 3/3/1) and ASF/Lip/pDNA (1/3/1) complexes significantly reduced *PLK1* expression compared to Lip/pDNA (0/3/1) complexes (Figure 4D). Correspondingly, CASF/Lip/pDNA (1/3/1, 3/3/1) and ASF/Lip/pDNA (1/3/1) complexes significantly promoted A549 cell apoptosis while inhibiting A549 cell proliferation compared with Lip/pDNA (0/3/1) complexes (Figure 5A–C). After transfection of the CASF/Lip/pDNA (1/3/1) complex, the phenomenon of cells becoming round was the most obvious, which also verified that the CASF/Lip/pDNA (1/3/1) complex had the best tumor cell inhibition ability (Figure 3A). The fusion of CASF with liposomes improved the transfection efficiency of the Lip/pDNA binary complex, reduced the expression of PLK1 protein, and improved the inhibitory ability of the Lip/pDNA complex on tumor cells. In contrast, the inhibitory effect of the control complex ASF/Lip/pDNA (3/3/1) on A549 cells was not significantly different from that of the binary complex, indicating that CASF was more able to fuse with liposomes and achieve better tumor inhibition.

CASF/Lip/pDNA showed strong tumor cell inhibitory activity in vitro, with an inhibition rate of nearly 90% against lung cancer cell A549 within 72 h (Figure 4C). CRISPR/Cas9 gene editing system has shown effective anti-tumor ability in vivo. It has been reported that the survival rate of tumor-bearing mice was increased to 60% within 60 days after treatment with the CRISPR/Cas9 gene editing system [59]. Therefore, it is speculated that CASF/Lip/pDNA should also have significant anti-tumor effects in vivo, which needs to be further verified by animal experiments.

CASF reduced the toxicity of Lip/pDNA complexes to normal lung cells by neutralizing the excess positive charge of liposomes. High charge density is one of the main reasons for the cytotoxicity of liposomes [60,61]. The surface zeta potential of the CASF/Lip/pDNA ternary complex was significantly lower than that of the Lip/pDNA binary complex (Figure 2B). The cell viability assay results showed that BEAS2B cells had higher viability after 24 h of incubation with CASF/Lip/pDNA than with Lip/pDNA complexes (Figure 3B). Correspondingly, cells cocultured with CASF/Lip/pDNA complexes had a more spreading morphology and smaller intercellular spaces (Figure 5A). The zeta potential and cell viability of the ASF/Lip/pDNA complex were generally consistent with those of the CASF/Lip/pDNA ternary complex, verifying that CASF reduces the cytotoxicity of liposomes by neutralizing the excess positive charge of liposomes.

Drugs must have an effective therapeutic effect and reliable safety to be used in clinical treatment [62]. The cytotoxicity of cationic liposomes is mainly due to their surface charge; however, reducing the surface charge often leads to a decrease in transfection efficiency. For example, modification of cationic liposomes by negatively charged hyaluronic acid can reduce the cytotoxicity of liposomes [63]. Unfortunately, the negatively charged molecules are not conducive to the interaction with the cell membrane, which may have an adverse effect on the transfection efficiency. The presence of RGD sequence in the H-chain of *Antheraea pernyi* silk fibroin can enhance the interaction between CASF/Lip/pDNA complex and cell membrane, and can shield part of the positive charge of cationic liposomes. Some cationic polymers, such as high-molecular-weight polyethylenimide, have also shown effective gene transfection in vitro when used as gene delivery vectors [64]. However, many cationic polymers are not biodegradable and have a negative impact on biocompatibility. In contrast, CASF is biodegradable and has good biocompatibility, which is beneficial for improving the cytocompatibility of liposomes. In addition, the method for CASF/Lip/pDNA complex preparation is simple, which facilitates mass production. CASF/Lip/pDNA complexes were prepared by incubating the mixture of CASF and Lip/pDNA for 15 min at room temperature.

The results of this study showed that compared with the Lip/pDNA binary complex, the transfection efficiency of CASF/Lip/pDNA ternary complex was improved in the presence of serum, showing the potential of cholesterolated silk fibroin-modified liposomes for in vivo application. However, there are still some issues to be further investigated. For example, whether the CRISPR-Cas9 gene editing system has off-target effects remains to be tested. Targeted delivery can increase the delivery efficiency to target cells and reduce the toxicity to normal tissues [65]. It is possible to further improve the effectiveness and safety of CASF/Lip/pDNA by modifying silk fibroin with lung cell-targeting molecules to give CASF/Lip/pDNA complexes the ability to specifically recognize and bind lung cancer cells. In addition, large-scale industrial production procedures and parameters to control the quality of CASF/Lip/pDNA ternary complexes need to be further designed and validated in order to facilitate the medical translation of complexes to industrial production and clinical applications.

## 5. Conclusions

FTIR and ^1^H-NMR results showed that thiolcholesterol was linked to the side chain of silk fibroin (ASF) by the bifunctional reagent SPDP to obtain cholesterol-modified silk fibroin (CASF). The zeta potential of the CASF/Lip/pDNA ternary complex was significantly lower than that of the Lip/pDNA binary complex. Cell transfection experiments showed that the CASF/Lip/pDNA ternary complex had higher transfection efficiency and a stronger *PLK1* gene knockout effect than the Lip/pDNA binary complex. Cell proliferation and apoptosis experiments showed that the CASF/Lip/pDNA ternary complex exhibited stronger lung cancer cell proliferation inhibition than the Lip/pDNA binary complex, and CASF/Lip/pDNA with a mass ratio of 1/3/1 showed the best lung cancer cell inhibition effect. The results of coculture with human bronchial epithelioid cells BEAS2B showed that the CASF/Lip/pDNA ternary complex was less toxic to the normal lung cell than the Lip/pDNA binary complex. Compared with the ASF/Lip/pDNA complex, the CASF/Lip/pDNA ternary complex has a higher transfection efficiency and stronger lung cancer cell inhibition effect, indicating that the silk fibroin modified by cholesterol can fuse with liposomes more effectively and achieve higher gene transfection efficiency and lung cancer cell inhibition effect. The above results indicate that CASF modification can effectively reduce the cytotoxicity of Lip/pDNA complexes while effectively improving the transfection efficiency, and thereby increasing the application potential of liposomes for lung cancer gene therapy.

## Figures and Tables

**Figure 1 pharmaceutics-15-02756-f001:**
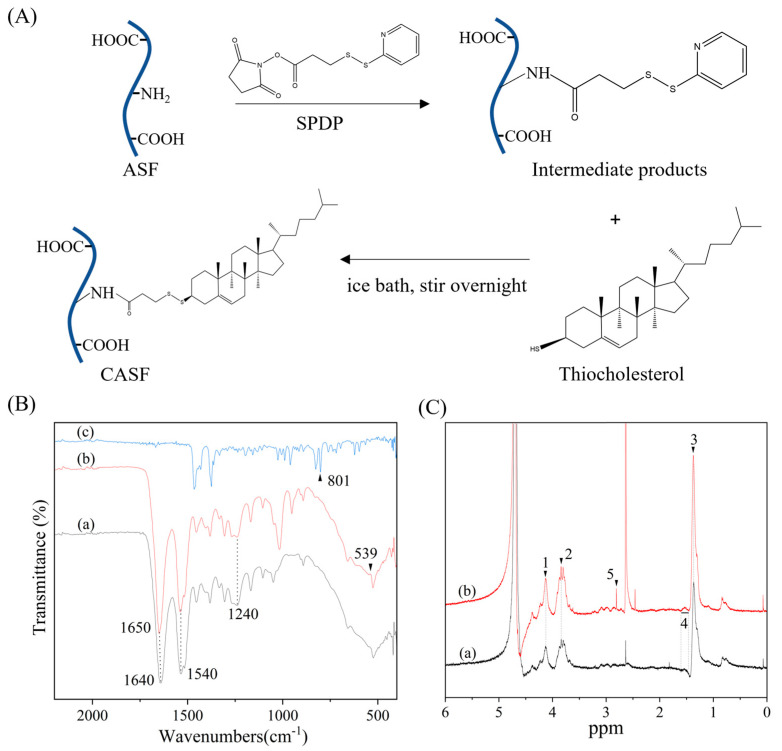
Preparation and characterization of CASF. (**A**) Schematic presentation of the synthetic reaction of CASF. (**B**) FTIR spectra. (a), (b) and (c) represent ASF, CASF and thiocholesterol, respectively. (**C**) ^1^H-NMR spectra. (a) and (b) denote ASF and CASF. 1: ~4.11 ppm; 2: ~3.84 ppm; 3: ~1.32 ppm; 4: 1.46~1.57 ppm; 5: ~2.75 ppm.

**Figure 2 pharmaceutics-15-02756-f002:**
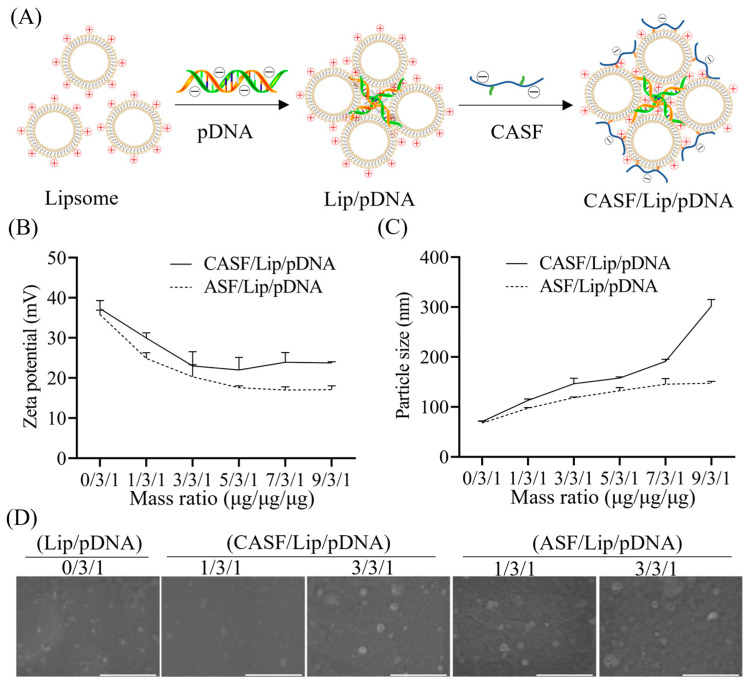
Preparation and characterization of CASF/Lip/pDNA complexes. (**A**) Schematic illustration of the formation of CASF/Lip/pDNA complexes. (**B**) Zeta potential and (**C**) particle size of CASF/Lip/pDNA and ASF/Lip/pDNA complexes with different mass ratios. The Lip/pDNA complex is represented as 0/3/1. (**D**) SEM images of Lip/pDNA (0/3/1), CASF/Lip/pDNA complexes with mass ratios of 1/3/1 and 3/3/1, and ASF/Lip/pDNA complexes with mass ratios of 1/3/1 and 3/3/1. Scale bar: 500 nm.

**Figure 3 pharmaceutics-15-02756-f003:**
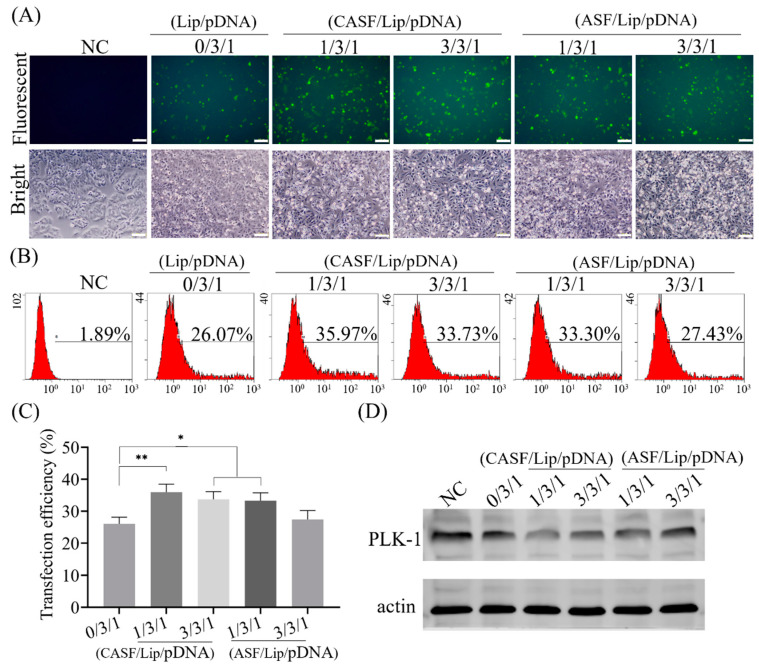
The transfection efficiency and PLK1 knockout effect of Lip/pDNA (0/3/1), CASF/Lip/pDNA (1/3/1, 3/3/1) and ASF/Lip/pDNA (1/3/1, 3/3/1) complexes in A549 cells. (**A**) Fluorescence microscopy images of A549 cells transfected with different complexes after 24 h, scale bar: 200 μm. (**B**) Flow cytometry images and (**C**) transfection efficiency histograms 24 h after transfection. (**D**) The expression of the PLK1 gene was analyzed by Western blotting 48 h after transfection. Untransfected cells were used as negative controls (NC). *: *p* < 0.05; **: *p* < 0.01.

**Figure 4 pharmaceutics-15-02756-f004:**
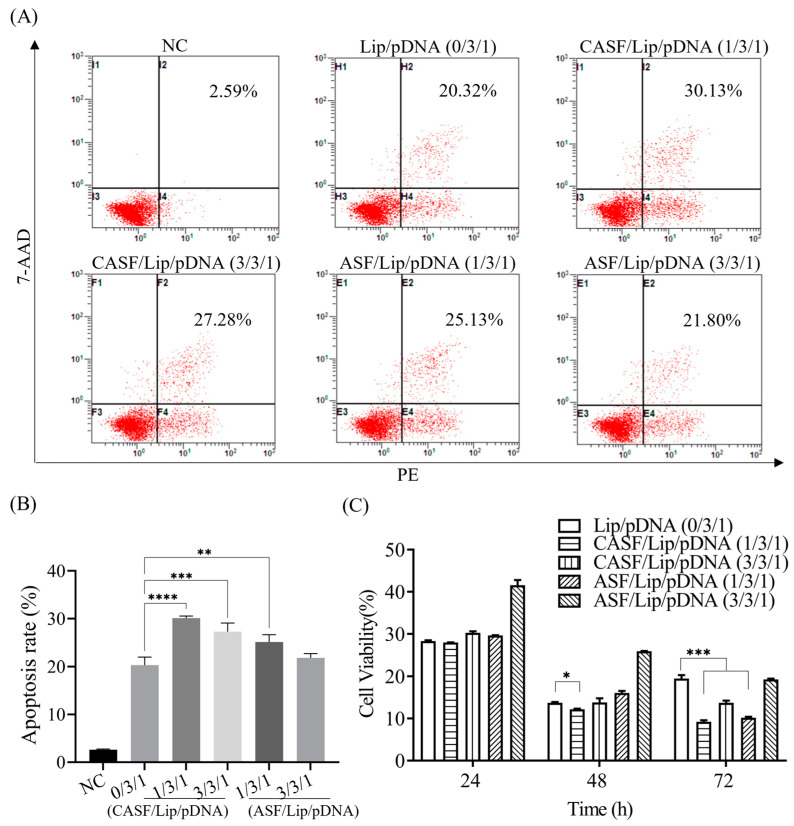
Apoptosis rate and proliferative activity of A549 cells transfected with Lip/pDNA (0/3/1), CASF/Lip/pDNA (1/3/1, 3/3/1) and ASF/Lip/pDNA (1/3/1, 3/3/1) complexes. (**A**) Flow cytometry images and (**B**) the corresponding histograms of the apoptotic rate 48 h after transfection. Untransfected cells were used as negative controls (NC). (**C**) Cell viability at 24, 48 and 72 h after transfection. *: *p* < 0.05; **: *p* < 0.01; ***: *p* < 0.001; ****: *p* < 0.0001.

**Figure 5 pharmaceutics-15-02756-f005:**
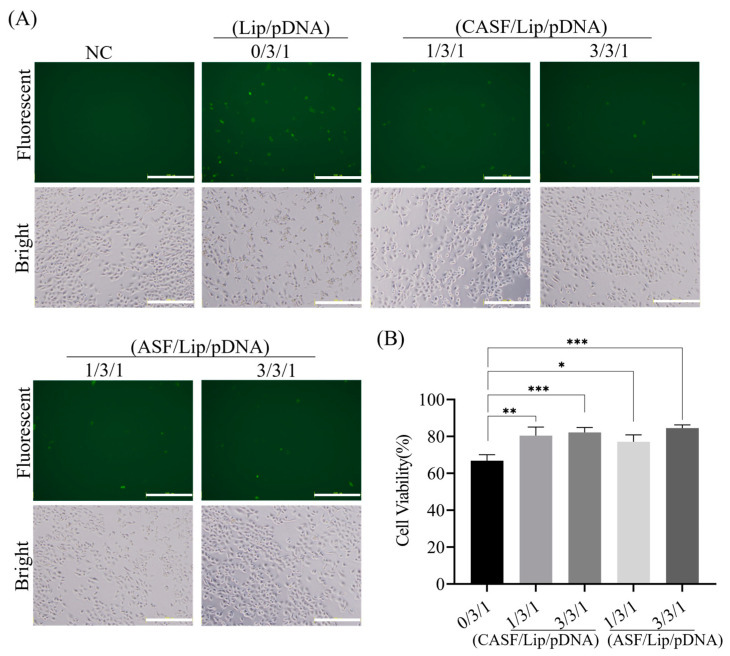
Green fluorescent protein expression and cell viability of BEAS2B cells at 24 h after coculture with Lip/pDNA (0/3/1), CASF/Lip/pDNA (1/3/1, 3/3/1) and ASF/Lip/pDNA (1/3/1, 3/3/1) complexes. (**A**) Fluorescence microscopy images. Cells that were not cocultured with the complex were used as a negative control (NC). Scale bar: 200 μm. (**B**) Cell viability. *: *p* < 0.05; **: *p* < 0.01; ***: *p* < 0.001.

## Data Availability

The data that support the findings of this study are available from the corresponding author upon reasonable request.

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
