# Peer review of "Silk Fibroin-Modified Liposome/Gene Editing System Knocks out the PLK1 Gene to Suppress the Growth of Lung Cancer Cells"

_pharmaceutics, 2023, doi:10.3390/pharmaceutics15122756_

Round 1

Reviewer 1 Report

Comments and Suggestions for Authors

The following study focused on the formation of pDNA complexes with liposomes and modified silk fibroin. The introduction provides a good background for understanding the course of the research. Materials and methods are described in a way that is understandable and can be reproduced. Most of the results are presented in a clear and legible way. The discussion of the results could be slightly extended and supplemented with a comparison of the effectiveness of the obtained carriers with others described in the literature. In my opinion, it would also be valuable to indicate potential problems related to the developed system and to describe further stages of research necessary to fully evaluate it.

Minor comments:

·        Section 2.2. – please indicate MWCO of the membrane used for dialysis

·        Section 2.3. - please indicate the resolution at which FTIR measurements were taken

·        Line 196 – upper index is missing in cell number per well

·        Section 2.11. – please indicate if DMEM used for BEAS-2B culture was supplemented with FBS or plates treated in any way to support cell adhesion

·        Line 287 – please indicate how those measurements of complexes on SEM images were done. How many measurements were taken for each sample?

·        Figure 3 – please correct scale bars on microscopy images, as now it is barely visible. I believe it would be beneficial to enlarge the microscopy images, as in the present form they are very difficult to analyse

·        Figure 3 and 4 – please keep the labelling of statistically significant differences consistent throughout the bar charts – either decide to label only statistically significant differences (and then skip “ns” labels) or provide information on all compared pairs (e.g., fig 4B – for me it looks like there should be statistically significant difference between NC and all other samples, which is not clearly labelled, whereas the insignificant difference between CASF/Lip/pDNA 0/3/1 and ASF/Lip/pDNA 3/3/1 is marked in the figure). Also it looks like some differences in fig. 4B are marked, whereas the others are not.

Author Response

1) The following study focused on the formation of pDNA complexes with liposomes and modified silk fibroin. The introduction provides a good background for understanding the course of the research. Materials and methods are described in a way that is understandable and can be reproduced. Most of the results are presented in a clear and legible way. The discussion of the results could be slightly extended and supplemented with a comparison of the effectiveness of the obtained carriers with others described in the literature. In my opinion, it would also be valuable to indicate potential problems related to the developed system and to describe further stages of research necessary to fully evaluate it.

Response: Thank you so much for your comments. According to your comments, we expanded the discussion section, added the comparison of effectiveness with other carriers, and the potential problems and needed further research are discussed (page 13 line 504 to page 14 line 533).

2) Section 2.2. – please indicate MWCO of the membrane used for dialysis.

Response: According to your comments, we added the MWCO of the membrane used for dialysis (page 3 line 121).

3) Section 2.3. - please indicate the resolution at which FTIR measurements were taken.

Response: According to your comments, we added the resolution adopted for FTIR measurement (page 3 line 127).

4) Line 196 – upper index is missing in cell number per well.

Response: According to your comments, we corrected the error (page 5 line 199).

5) Section 2.11. – please indicate if DMEM used for BEAS-2B culture was supplemented with FBS or plates treated in any way to support cell adhesion.

Response: DMEM added 10% FBS, did not add the other ingredients. The culture plates were ordinary polystyrene plates without any other treatment. According to your comments, we added the culture conditions for BEAS-2B (page 5 line 209 to line 210).

6) Line 287 – please indicate how those measurements of complexes on SEM images were done. How many measurements were taken for each sample?

Response: The particle size of the complexes in SEM images was measured using Image-J (1.51J8) software, at least 100 complexes were measured in each group of samples. We added the measurement details in page 4 line 167 to line 168.

7) Figure 3 – please correct scale bars on microscopy images, as now it is barely visible. I believe it would be beneficial to enlarge the microscopy images, as in the present form they are very difficult to analyses.

Response: According to your comments, we correct the scale bars on the microscope images in Figure 3 and Figure 5.

8) Figure 3 and 4 – please keep the labelling of statistically significant differences consistent throughout the bar charts – either decide to label only statistically significant differences (and then skip “ns” labels) or provide information on all compared pairs (e.g., fig 4B – for me it looks like there should be statistically significant difference between NC and all other samples, which is not clearly labelled, whereas the insignificant difference between CASF/Lip/pDNA 0/3/1 and ASF/Lip/pDNA 3/3/1 is marked in the figure). Also it looks like some differences in fig. 4B are marked, whereas the others are not.

Response: According to your comments, we only marked statistically significant differences and removed the "NS" labels in Figure 3 and Figure 4.

Reviewer 2 Report

Comments and Suggestions for Authors

This study investigating the use of a CRISPR‒Cas9 gene knockout system for targeting PLK1 in lung cancer cells is certainly promising. However, there are some points that might benefit from further clarification or improvement.

  1. Vector Suitability for In Vivo Delivery: The identification of a vector suitable for in vivo delivery is a critical aspect for clinical translation. While the study highlights the effectiveness of the CASF/Lip/pDNA ternary complex in vitro, the discussion around its potential in vivo application seems limited. Have there been any initial in vivo trials or considerations for future animal studies to assess its efficacy and safety in a more physiological setting?
  2. Mechanism of Action and Off-Target Effects: Understanding the mechanism behind the observed effects would greatly strengthen the paper. Have the off-target effects of the CRISPR‒Cas9 system been assessed comprehensively? Addressing this aspect could enhance the reliability and safety of the proposed therapeutic approach.
  3. References and Comparative Analysis: Including more references discussing similar approaches or studies in the field would add depth to the discussion. Comparing the findings of this study with others in the literature could strengthen the paper's conclusions. Please use: https://doi.org/10.3892/or.2019.6964, https://doi.org/10.3892/or.2018.6795, https://doi.org/10.3390/pr9040621
  4. Clinical Translation and Limitations: While the paper discusses the potential for clinical application, addressing potential limitations or challenges in transitioning this approach to clinical settings would be insightful. For instance, what challenges might arise during large-scale production or regulatory considerations?
  5. Long-term Effects: Assessing the long-term effects of PLK1 knockout and the sustained efficacy of the CASF/Lip/pDNA complex over extended periods could provide a more comprehensive understanding of its therapeutic potential.
  6. Verification of Results: It might be beneficial to verify the results using alternative methods or assays to confirm the observed effects on cell proliferation, apoptosis, and cytotoxicity.

Comments on the Quality of English Language

This study investigating the use of a CRISPR‒Cas9 gene knockout system for targeting PLK1 in lung cancer cells is certainly promising. However, there are some points that might benefit from further clarification or improvement.

  1. Vector Suitability for In Vivo Delivery: The identification of a vector suitable for in vivo delivery is a critical aspect for clinical translation. While the study highlights the effectiveness of the CASF/Lip/pDNA ternary complex in vitro, the discussion around its potential in vivo application seems limited. Have there been any initial in vivo trials or considerations for future animal studies to assess its efficacy and safety in a more physiological setting?
  2. Mechanism of Action and Off-Target Effects: Understanding the mechanism behind the observed effects would greatly strengthen the paper. Have the off-target effects of the CRISPR‒Cas9 system been assessed comprehensively? Addressing this aspect could enhance the reliability and safety of the proposed therapeutic approach.
  3. References and Comparative Analysis: Including more references discussing similar approaches or studies in the field would add depth to the discussion. Comparing the findings of this study with others in the literature could strengthen the paper's conclusions. Please use: https://doi.org/10.3892/or.2019.6964, https://doi.org/10.3892/or.2018.6795, https://doi.org/10.3390/pr9040621
  4. Clinical Translation and Limitations: While the paper discusses the potential for clinical application, addressing potential limitations or challenges in transitioning this approach to clinical settings would be insightful. For instance, what challenges might arise during large-scale production or regulatory considerations?
  5. Long-term Effects: Assessing the long-term effects of PLK1 knockout and the sustained efficacy of the CASF/Lip/pDNA complex over extended periods could provide a more comprehensive understanding of its therapeutic potential.
  6. Verification of Results: It might be beneficial to verify the results using alternative methods or assays to confirm the observed effects on cell proliferation, apoptosis, and cytotoxicity.

Author Response

This study investigating the use of a CRISPR‒Cas9 gene knockout system for targeting PLK1 in lung cancer cells is certainly promising. However, there are some points that might benefit from further clarification or improvement.

1) Vector Suitability for In Vivo Delivery: The identification of a vector suitable for in vivo delivery is a critical aspect for clinical translation. While the study highlights the effectiveness of the CASF/Lip/pDNA ternary complex in vitro, the discussion around its potential in vivo application seems limited. Have there been any initial in vivo trials or considerations for future animal studies to assess its efficacy and safety in a more physiological setting?

Response: Thank you for your comments. We are currently conducting animal experiments of CASF/Lip/pDNA ternary complex for the treatment of lung cancer xenograft in mice and the acute systemic toxicity experiment in mice. According to the preliminary experimental results, the CASF/Lip/pDNA ternary complex has obvious inhibitory effect on lung cancer xenograft and reduces the expression of PLK1 in vivo, and has no obvious systemic acute toxicity. Since the animal experiments have not yet been completed and the amount of data is large, we consider publishing it in another article in the future. This paper focuses on the construction, characterization and in vitro biological effects of CASF/Lip/pDNA complexes. According to your comments, we have added a discussion of the potential of CASF/Lip/pDNA ternary complexes for in vivo applications (page 13 line 486 to line 492).

2) Mechanism of Action and Off-Target Effects: Understanding the mechanism behind the observed effects would greatly strengthen the paper. Have the off-target effects of the CRISPR‒Cas9 system been assessed comprehensively? Addressing this aspect could enhance the reliability and safety of the proposed therapeutic approach.

Response: The knockout of proto-oncogene PLK1 by CRISPR-Cas9 gene editing system can effectively inhibit the proliferation of tumor cells. CASF modified liposomes can effectively improve the transfection efficiency of liposomes, thereby improving the ability to inhibit the proliferation of tumor cells. CRISPR-Cas9 gene editing system is less damaging to the genome when used for gene knockout than gene editing systems such as zinc finger nucleases (Shen B, et al. Nat Methods. 2014 Apr;11(4):399-402.). Secondly, we preferentially selected the sequence with the lowest off-target efficiency when designing the sgRNA sequence, so we did not comprehensively evaluate the off-target effect of the CRISPR-Cas9 system. However, evaluation of the off-target effects of CRISPR-Cas9 gene editing system is helpful to improve the safety of treatment. According to your comments, we added a discussion about the off-target effects of the CASF/Lip/pDNA complex (page 13 line 521 to page 14 line 530).

3) References and Comparative Analysis: Including more references discussing similar approaches or studies in the field would add depth to the discussion. Comparing the findings of this study with others in the literature could strengthen the paper's conclusions. Please use:

https://doi.org/10.3892/or.2019.6964, https://doi.org/10.3892/or.2018.6795, https://doi.org/10.3390/pr9040621

Response: According to your comments, we have added comparisons with similar methods or studies in this field and supplemented references (page 12 line 419 to line 422; page 13 line 504 to line 520).

4) Clinical Translation and Limitations: While the paper discusses the potential for clinical application, addressing potential limitations or challenges in transitioning this approach to clinical settings would be insightful. For instance, what challenges might arise during large-scale production or regulatory considerations?

Response: We consider that there may be two main issues that restrict the medical translation of CASF/Lip/pDNA ternary complexes. First, the targeting of the complex to lung cancer cells needs to be further enhanced; The second is how to control the quality of the complex in industrial production. According to your comments, we have added a discussion of this issue to the revised manuscript. (page 13 line 521 to page 14 line 533).

5) Long-term Effects: Assessing the long-term effects of PLK1 knockout and the sustained efficacy of the CASF/Lip/pDNA complex over extended periods could provide a more comprehensive understanding of its therapeutic potential.

Response: According to your comments, we have added to the discussion of the long-term effects of PLK1 knockdown and the sustained efficacy of the CASF/Lip/pDNA complex over extended periods (page 13 line 486 to line 492).

6) Verification of Results: It might be beneficial to verify the results using alternative methods or assays to confirm the observed effects on cell proliferation, apoptosis, and cytotoxicity.

Response: In the original paper, we evaluated the effects of CASF/Lip/pDNA complexes on cell proliferation, apoptosis, and cytotoxicity using confocal laser microscopy images, apoptosis rate, and cell viability, respectively. According to your comments, we expanded the observation, description, and discussion of cell morphology in the bright-field images to further demonstrate the effects of the complex on cell proliferation, apoptosis, and cytotoxicity in the revised manuscript, (page 8 line 313 to line 322; page 11 line 394 to line 401, page 13 line 477 to line 480, line 499 to line 500).

Reviewer 3 Report

Comments and Suggestions for Authors

At the center of this paper is the design, synthesis, and testing of a ternary complex CASF/Lip/pDNA consisting of positively charged liposomes around pDNA, and negatively charged CASF to hold the liposomes in place by ameliorating their positive charge. This paper presents a rational set of experiments designed to test the veracity of the proposal that the complex is transfected into lung cancer cells A549, to assess the transfection efficiency, the PLK1 gene knockout effect, and the inhibitory effect on lung cancer cell proliferation. By showing that the complex is transfected with significantly high efficiency, reducing the PLK1 cancer cell expression and proliferation, while increasing apoptotic cell death, the paper shows that this complex represents a technological advancement in the delivery of the CRISPR‒Cas9 system, and that this can effectively inhibit the proliferation of tumor cells.

The chemical conjugations are adequately supported by the FTIR and NMR experiments which prove that the N-succinimidyl-3-(2-pyridyldithio)propionate (SPDP) was successfully conjugated by amide bonds, and that the thio-cholesterol was linked to the side chain of Antheraea pernyi silk fibroin (ASF), as planned. Although the observation of the ASF/Lip/pDNA nano-complexes under the scanning electron microscope and the measurement of their particle sizes proves formation of the nanostructures, the internal structure proposed in Figure 2(A) was not elucidated.

Nevertheless, measurements of the transfection, and efficiency thereof, the PLK1 knockout effect of Lip/pDNA, and CASF/Lip/pDNA, the apoptosis and proliferative effect on A549 cells, the fluorescent protein expression and viability of BEAS2B cells, were all described in detail. Therefore the importance of the internal structure and architecture of the nanoconjugates may be academic.

Author Response

At the center of this paper is the design, synthesis, and testing of a ternary complex CASF/Lip/pDNA consisting of positively charged liposomes around pDNA, and negatively charged CASF to hold the liposomes in place by ameliorating their positive charge. This paper presents a rational set of experiments designed to test the veracity of the proposal that the complex is transfected into lung cancer cells A549, to assess the transfection efficiency, the PLK1 gene knockout effect, and the inhibitory effect on lung cancer cell proliferation. By showing that the complex is transfected with significantly high efficiency, reducing the PLK1 cancer cell expression and proliferation, while increasing apoptotic cell death, the paper shows that this complex represents a technological advancement in the delivery of the CRISPR‒Cas9 system, and that this can effectively inhibit the proliferation of tumor cells.

1) The chemical conjugations are adequately supported by the FTIR and NMR experiments which prove that the N-succinimidyl-3-(2-pyridyldithio)propionate (SPDP) was successfully conjugated by amide bonds, and that the thio-cholesterol was linked to the side chain of Antheraea pernyi silk fibroin (ASF), as planned. Although the observation of the ASF/Lip/pDNA nano-complexes under the scanning electron microscope and the measurement of their particle sizes proves formation of the nanostructures, the internal structure proposed in Figure 2(A) was not elucidated.

Response: Thank you so much for your comments. Small granular liposomes compress pDNA to form larger aggregates (Lip/pDNA), with pDNA inside and liposomes outside (Dan, N., et al. Adv. Colloid. Interfac. 2014, 205: 230–239). CASF fuses with lipids on the surface of the Lip/pDNA complex by hydrophobic interaction forces to form the CASF/Lip/pDNA complex. We characterized the surface morphology and particle size of the complexes by scanning electron microscopy and particle size analyzer. The results showed that the surface morphology and size of the CASF/Lip/pDNA complex favored cell transfection. As you mentioned, observation or testing techniques such as TEM can better characterize the internal structure of the complex, which needs more detailed work in the future. According to your comments, we have improved the description of the internal structure of the CASF/Lip/pDNA complex (page 6, lines 258 to 259).

2) Nevertheless, measurements of the transfection, and efficiency thereof, the PLK1 knockout effect of Lip/pDNA, and CASF/Lip/pDNA, the apoptosis and proliferative effect on A549 cells, the fluorescent protein expression and viability of BEAS2B cells, were all described in detail. Therefore, the importance of the internal structure and architecture of the nanoconjugates may be academic.

Response: As you pointed out, elucidating the internal structure of the CASF/Lip/pDNA complex is very academic. In this paper, we speculated the structure of the complex according to the results of the literatures, and then characterized it by scanning electron microscopy and particle size analyzer. We consider the use of TEM in our follow-up work and the establishment of layer-by-layer dissociation techniques on the surface of CASF/Lip/pDNA complexes to investigate the details of the internal structure of CASF/Lip/pDNA complexes. According to your comments, we have improved the description of the internal structure of the CASF/Lip/pDNA complex (page 6, lines 258 to 259).

Reviewer 4 Report

Comments and Suggestions for Authors

In the present manuscript the authors report a bio-supramolecular complex among DNA/ cationic liposome/cholesterol modified silk fibroin able to knock out PLK1 gene in cancer cell. The introduction is clear and provides a solid justification for the work. The methodology protocols are described in detail while the experimental data are critically analyzed. Finally, the quality of the figures is of a high level and the number of cited works is appropriate. The overall work appears to be solid and complete, nevertheless a few points need to be better clarified before recommending publication.

-          Did the author evaluate the average number of reactive sites on ASF able to couple with thiocholesterol?

-          The reaction of fibroin with SPDP might lead to undesired inter or intramolecular further sulfide exchange with free amino acids like cys creating protein clusters. How did the author avoid unspecific reactions?

-          IR and 1H-NMR are informative characterization techniques but hardly sufficient to clearly characterize macromolecules or determine the amount of tagged site. MALDI-TOF could be more informative to get this information. Did the author consider performing such an analysis?

-          Both the IR and NMR spectra might be more informative if completed with the spectra of the “intermediate product” focusing on the Pyridine signals which usually can be clearly identified both in IR and NMR.

-          It is hard to understand how in the NMR spectra the authors can clearly identify the NH signals in D2O but visually it is very hard to see the cholesterol CH2 signals which should be found in the aliphatic region of the spectrum.

Author Response

In the present manuscript the authors report a bio-supramolecular complex among DNA/ cationic liposome/cholesterol modified silk fibroin able to knock out PLK1 gene in cancer cell. The introduction is clear and provides a solid justification for the work. The methodology protocols are described in detail while the experimental data are critically analyzed. Finally, the quality of the figures is of a high level and the number of cited works is appropriate. The overall work appears to be solid and complete, nevertheless a few points need to be better clarified before recommending publication.

1) Did the author evaluate the average number of reactive sites on ASF able to couple with thiocholesterol?

Response: Thank you so much for your comments. We evaluated the average number of ASF-reactive sites capable of coupling to thiocholesterol. Thiocholesterol is coupled to the amino group on the side chain of ASF mediated by SPDP. The contents of Arg, His and Lys in Antheraea pernyi silk fibroin are approximately 2.59%, 0.80% and 0.07% respectively. According to your comments, we have added a discussion of ASF reactivity sites (page 12 line 438 to line 441).

2) The reaction of fibroin with SPDP might lead to undesired inter or intramolecular further sulfide exchange with free amino acids like cys creating protein clusters. How did the author avoid unspecific reactions?

Response: Since regenerated Antheraea pernyi silk fibroin basically does not contain Cys or other molecules containing free sulfhydryl groups (Silva SS, et al. Chinese Oak Tasar Silkworm Antheraea pernyi Silk Proteins:Current Strategies and Future Perspectives for Biomedical Applications. Macromol Biosci. 2019, 19(3): e1800252), the possibility of non-specific reaction is small, so we did not take measures to prevent non-specific reactions.

In consideration of your concerns, we have added a description of this issue to the discussion section of the revised manuscript (page 12 line 441 to line 442).

3) IR and 1H-NMR are informative characterization techniques but hardly sufficient to clearly characterize macromolecules or determine the amount of tagged site. MALDI-TOF could be more informative to get this information. Did the author consider performing such an analysis?

Response: IR and 1H-NMR spectra basically confirmed that cholesterol was linked to the silk fibroin side chain via disulfide bonds, so MALDI-TOF was not used. Your suggestion is very reasonable and more information can be obtained by MALDI-TOF. According to your comments, we performed further analysis of the IR and 1H-NMR results (page 5 line 229 to line 230, line 234).

4) Both the IR and NMR spectra might be more informative if completed with the spectra of the “intermediate product” focusing on the Pyridine signals which usually can be clearly identified both in IR and NMR.

Response: As you pointed out, the IR and NMR spectra of "intermediate products" can provide more information. However, the IR and NMR spectra can basically prove that cholesterol is coupled to the side chain of silk fibroin through a disulfide bond, so the IR and NMR spectra of "intermediate products" were not examined.

5) It is hard to understand how in the NMR spectra the authors can clearly identify the NH signals in D2O but visually it is very hard to see the cholesterol CH2 signals which should be found in the aliphatic region of the spectrum.

Response: According to the literatures, the proton peak near the 4.13 region is the proton peak in -NH-C(=O)- (Zhang, B., et al. Lung cancer gene therapy: Transferrin and Hyaluronic Acid Dual Ligand-Decorated Novel Lipid Carriers for Targeted Gene Delivery. Oncol. Rep. 2017, 37: 937–944), so by comparing the structure and spectrum, we analyzed that this is the -NH- signal. The chemical shift of the proton of -CH2- in cholesterol appears in the range of 1.46 to 1.57 ppm. According to your comments, we have added the description of the cholesterol -CH2- signal (page 5 line 242 to page 6 line 244).

Round 2

Reviewer 2 Report

Comments and Suggestions for Authors

Accept

Reviewer 4 Report

Comments and Suggestions for Authors

In the revised version of the manuscript the authors have improved the manuscript and addressed the reviewer comments. I have no additional points to highlight therefore I do reccomend publication in the present form.